
# Dust impact on surface solar irradiance assessed with model simulations, satellite observations and ground-based measurements

Panagiotis G. Kosmopoulos[1,2], Stelios Kazadzis[3], Michael Taylor[2], Eleni Athanasopoulou[1], Orestis Speyer[1], Panagiotis I. Raptis[1], Eleni Marinou[2,4], Emmanouil Proestakis[4,5], Stavros Solomos[4], Evangelos Gerasopoulos[1,6], Vassilis Amiridis[4], Alkiviadis Bais[2], Charalabos Kontoes[4]

[1]Institute for Environmental Research and Sustainable Development, National Observatory of Athens, Greece
[2]Laboratory of Atmospheric Physics, Aristotle University of Thessaloniki, Greece
[3]Physicalisch-Meteorologisches Observatorium Davos, World Radiation Center, Switzerland
[4]Institute for Astronomy, Astrophysics, Space Applications and Remote Sensing, National Observatory of Athens, Greece
[5]Laboratory of Atmospheric Physics, Department of Physics, University of Patras, Greece
[6]Navarino Environmental Observatory, Messenia, Greece

*Correspondence to:* P.G. Kosmopoulos (pkosmo@meteo.noa.gr)

**Abstract.** This study assesses the impact of dust on surface solar radiation focussing on an extreme dust event. For this purpose, we exploited the synergy of AERONET measurements and passive and active satellite remote sensing (MODIS and CALIPSO) observations, in conjunction with radiative transfer model (RTM) and chemical transport model (CTM) simulations and the 1-day ahead forecasts from the Copernicus Atmosphere Monitoring Service (CAMS). The area of interest is the eastern Mediterranean where anomalously high aerosol loads were recorded between the 30 January and 3 February 2015. The intensity of the event was extremely high, with aerosol optical depth (AOD) reaching 3.5, and optical/microphysical properties suggesting aged dust. RTM and CTM simulations were able to quantify the extent of dust impact on surface irradiances and reveal substantial reduction in solar energy exploitation capacity of PV and CSP installations, under this high aerosol load. We found that such an extreme dust event can result to Global Horizontal Irradiance (GHI) attenuation by as much as 40-50%, a much stronger Direct Normal Irradiance (DNI) decrease (80-90%), while spectrally this attenuation is distributed to 37% in the UV region, 33% to the visible and around 30% to the infrared. CAMS forecasts provided a reliable available energy assessment (accuracy within 10% of that obtained from MODIS). Spatially, the dust plume resulted in a zonally-averaged reduction of GHI and DNI of the order of 150W/m$^2$ in southern Greece, and a mean increase of 20 W/m$^2$ in the northern Greece as a result of lower AOD values combined with local atmospheric processes. This analysis of a real-world scenario contributes to the understanding and quantification of impact range of high aerosol loads on solar energy and the potential for forecasting power generation failures at sunshine-privileged locations where solar power plants exist, are under construction, or being planned.

**Keywords.** Dust; Solar Radiation; Atmospheric Aerosol; Radiative Transfer; AERONET; MODIS; CALIPSO; COSMO-ART; CAMS





## 1 Introduction

Solar energy potential is sensitive to various atmospheric parameters. In addition to the solar zenith angle (SZA) as the key determining factor, cloud presence is another factor that attenuates solar radiation arriving at the Earth's surface. For this, large photovoltaic (PV) installations are built where solar insolation is high and cloud-free sky conditions prevail for the largest part of the year. However, in the case of European sponsored installations, these are also significantly affected by mineral dust (e.g. the 160 MW Noor Concentrated Solar Power (CSP) in Morocco).

The aerosol radiative effects over the eastern Mediterranean have been studied systematically in the last decade (e.g. Papadimas et al., 2012; Turnock et al., 2015; Lindfors et al., 2013). It is a semi-enclosed sea surrounded by continental area with discrete sources of aerosols and it is characterized by large direct radiative effects due to high and frequently dust loads, especially during spring (Kosmopoulos et al., 2008; Gkikas et al., 2012; 2013; Flaounas et al., 2015; Athanasopoulou et al., 2016). A recent climatology of global aerosol mixtures, derived from 7 years GOCART model simulations (Taylor et al., 2015), suggests that dust is the primary component of aerosol mixtures over the eastern Mediterranean. While the spatial extent of dust mixtures is fairly stable on the seasonal timescale, they are highly variable in time at the local scale. Desert dust plays an important role in the radiative forcing (RF), with an estimated Top of Atmosphere (TOA) RF in the range of -0.6 to 0.5 W/m$^2$ (IPCC, 2013). Still, dust induced RF is very uncertain in both magnitude and sign, an uncertainty driven by the chemical composition of mineral particles (Claquin et al., 1998), the wavelength dependence of their optical properties (e.g. single-scattering albedo -SSA-, asymmetry factor) as well as the albedo of the underlying surface and the relative height between the dust layer and the clouds (Kinne and Pueschel, 2001).

In the absence of clouds, dust is the main source of attenuation of the surface solar radiation (SSR), with the direct normal irradiance (DNI) being affected much more intensively than the global horizontal irradiance (GHI). Many studies focus on different approaches to quantify and estimate the impact of dust on the SSR (Dirnberger et al., 2015; Ishii et al., 2013; Lindfors et al., 2013; Allen et al., 2013; Qian et al., 2007; Papayiannis et al., 2005). In order to assess the impact of strong dust events on solar energy, a monthly climatology of aerosol optical depth (AOD) and spectrally-integrated SSR, including the direct normal irradiance (DNI) and the global horizontal irradiance (GHI), was calculated with the radiative transfer model (RTM) libRadtran. The present study computes the direct effect of the extreme dust event of the 1 February 2015 on the radiative energy budget using satellite and ground-based data as input to the RTM. The study area covers the eastern Mediterranean and more specifically the region of Greece. Dust cases in winter are rare, but intense when they occur (Gerasopoulos et al., 2011; Kosmopoulos et al., 2011), thus the study of such an event is of great interest with respect to low incoming SSR and the typical winter meteorological conditions. In order to better understand this effect, we used data from different sources to perform a multi-model analysis of the intense incursion of Saharan dust into South-Eastern Europe during the study period that began with cyclonogenesis to the North of Libya on 28/01/2015, peaked over the Mediterranean on the 01/02/2015 and dissipated a couple of days later. The data synergy is provided by i) RTM simulations from libRadtran, ii) chemical transport model (CTM) simulations from COSMO-ART, iii) satellite aerosol retrievals from the



MODIS spectroradiometer, iv) aerosol profiling from CALIPSO, v) aerosol retrievals provided by AERONET sunphotometers and their inversion algorithm and finally, vi) aerosol product retrievals from the CAMS. We measure the attenuation of SSR during the course of the evolution of the dust outbreak and correlate it with the available data (e.g. CAMS against MODIS) to assess the relative impact of dust aerosol on solar power over the study area. The work is

organized as follows: in Sect. 2 the measurements and models used are presented; then the results from the 3D observation and ground-based dimension of the dust event evolution, together with the impacts on surface solar radiation are presented in Sect. 3; finally, the conclusions are provided in Sect. 4.

## 2 Measurements & Models

### 2.1 Measurements

#### 2.1.1 MODIS and CALIPSO

The MODerate resolution Imaging Spectroradiometer (MODIS) sensors are among the primary instruments onboard the polar orbit sun-synchronous NASA satellites Terra and Aqua (Salomonson et al., 1989). MODIS sensors provide retrievals of AOD at 550nm since February 2000 and June 2002 for Terra and Aqua satellites respectively. The retrievals of MODIS sensors are established against different types of earth surface, based on a pair of complementary algorithms, the "Dark

Target" (DT) and "Deep Blue" (DB). The DT algorithm is used over vegetated/dark land surfaces which are characterized by low reflectance and additionally over ocean, while the DB algorithm provides AOD retrievals over bright and arid land surfaces (Levy et al., 2013). The accuracy of Collection 6 (C6) MODIS DT algorithm is approximately equal to $\pm$ (0.05 + 0.15$\tau_A$) and + (0.04 + 0.1 $\tau_A$), - (0.02 + 0.1 $\tau_A$) over land and oceanic surfaces respectively, while the expected error of DB algorithm is estimated at $\pm$ (0.03 + 0.2 $\tau_M$) (Levy et al., 2013; Sayer et al., 2015; Georgoulias et al., 2016).  The input

parameters $\tau_A$ and $\tau_M$ at the error estimation of the two algorithms correspond to the AOD derived by AERONET and MODIS respectively. The products of MODIS are provided in different levels of processing. The spatial resolution of MODIS Level 2 (L2) is approximately 10 km x 10 km at nadir viewing geometry, while grids are increasing significantly with increasing viewing angle. In this paper MODIS Aqua C6 L2 is used and in addition to the $AOD_{550nm}$ the Cloud Fraction (CF) over land, ocean and the DB CF over land are combined to provide the full information on the cloud coverage during

the Aqua overpass (Platnick et al., 2016).

For the vertical distribution and structure of the atmosphere the synergy of MODIS and CALIOP (Cloud Aerosol Lidar with Orthogonal Polarization) can provide a unique 3-dimensional characterization of dust outflows (Gkikas et al., 2016; Kosmopoulos et al. 2011). CALIOP, the main instrument onboard CALIPSO (Cloud-Aerosol Lidar and Infrared Pathfinder Satellite Observations), provides information on the vertical distribution of aerosols and clouds (Winker et al., 2009) since

June 2006. CALIPSO, the NASA-CNES collaboration project, is part of the A-Train constellation and therefore observes the same atmospheric layer as MODIS/Aqua at nadir viewing, with a delay of a few seconds only. CALIPSO main products



include the total attenuation backscatter coefficient, at both 532nm and 1064nm, and polarization retrievals at 532nm. Based on the backscatter coefficient, the depolarization ratio, the altitude of the atmospheric layers and the surface characteristics below the orbit of CALIPSO, the CALIOP algorithm (Omar et al., 2009; Omar et al., 2016) classifies the atmospheric masses among different feature types (clear air, cloud, aerosol, stratospheric feature, surface, subsurface or totally

attenuated). In the case of aerosols, the algorithm further discriminates the atmospheric layers between marine, dust, clean continental, polluted continental/smoke, polluted dust elevated smoke, dusty marine, PSC aerosol, volcanic ash and sulfate/other. In this work, the CALIPSO Level 2 Version 4.10 aerosol profile product is used.

### 2.1.2 AERONET

The AERONET measurements reported in this work were conducted at Thisio AERONET station (ATHENS-NOA), which

is located in the capital of Greece, Athens, with a CIMEL sunphotometer (CE318). The instrumentation, data acquisition, retrieval algorithms and calibration procedure conform with the standards of the AERONET and are described in detail in numerous studies (e.g. Holben et al., 2001; Dubovik et al., 2000). Typically, the total uncertainty in AOD for the field instrument under cloud-free conditions, is ±0.01 for λ>440 nm, and ±0.02 for shorter wavelengths. In this study, the hourly-averaged data, which are cloud-screened and quality assured (Smirnov et al., 2000), were used. The temporal resolution of

AERONET AOD measurements is very high (≈ 1 per 10 minutes).

### 2.2 Models

### 2.2.1 Meteorology and back-trajectories

The final analysis data (FNL) of the National Center for Environmental Protection (NCEP) are used for the assessment of the meteorological conditions related to the uplift and transport of dust. The NCEP-FNL (Final Analysis) data are on 1°×1° grid

and are available every six hours (NCEP 2000). The analyses include meteorological parameters (pressure, geopotential height, temperature, wind etc.) inside the boundary layer, at the surface and at 26 pressure levels from 1000 hPa to 10 hPa. The HYSPLIT dispersion model (Stein et al., 2015) is used for the computation of air mass back-trajectories during the dust episode. The trajectories are calculated from 500 – 2000 m every 500 m over the Aegean Sea in order to define the transport paths of dust originally elevated at the coastal sources of N. Africa. HYSPLIT model is driven by NCEP-GDAS

meteorological data at 1°×1° resolution (NCEP 2000).

### 2.2.2 COSMO-ART simulations

COSMO-ART is a regional atmospheric model which couples online meteorology and chemistry. COSMO is the operational numerical weather prediction model of the German and other European weather services (Baldauf et al., 2011) and is used as a regional climate model in a modified version CCLM (Rockel et al., 2008). ART (Aerosols and Reactive Trace gases) is the

chemistry extension of COSMO. Detailed description of the model, of the physico-chemical characteristics of the aerosol



modes and of the parameterizations of feedbacks of aerosols on radiation, temperature, cloud and ice condensation nuclei (CCN and IN) are given in Vogel et al. (2009), Bangert et al. (2011; 2012), and Rieger et al. (2014). The model domain used in this study is defined so that it includes the area of the dust source (NW Africa) and its transport path towards SE Mediterranean (from 24° to 42° N and from 5° W to 32° E). The horizontal spatial resolution is 0.25°, while the vertical

extent reaches 22.7 km, stratified in 40 layers. The meteorological initialization is performed using inputs from the ICON general circulation model (Zängl et al., 2015), i.e. from the operational model runs of the German Weather Service. Anthropogenic emissions are derived from the TNO-MACC III database (Kuenen et al., 2014), while the African dust emissions are calculated online, thus are case specific. Their hourly emission rate is parameterized according to the saltation processes, as a function of friction velocity (when greater than a threshold), soil water content, and surface roughness. More

information on the exact methodology can be found in Vogel et al. (2006). Apart from the base-case run, a scenario with the online interaction between dust and radiation switched off is also performed.

### 2.2.3 RTM simulations

For the RTM simulations we used libRadtran (Mayer & Kylling, 2005) in order to produce gridded GHI, DNI, VIS and UV spectral irradiances and integrated values at the surface with the impact of dust as well as for clean (aerosol-free) and clear

(cloud-free) sky conditions. The RTM simulations convert the satellite and ground-based cloud and aerosol products directly into high resolution (1nm) spectral irradiances (Emde et al., 2016). The main input is the AOD as to quantify the exact impact of the dust particles in local (Athens) and regional (Greece) level. Other basic input parameters to the RTM simulations were the SZA, total ozone, reflectivity of the earth's surface and water vapor column. Thus, the RTM has been applied to MODIS Level 3 AOD (550 nm) data with the spatial resolution of 1x1 degree and to CAMS 1-day ahead AOD

(550 nm) forecast to produce gridded spectra and spectrally-integrated total SSR values. The AERONET AOD data in the station of Athens were also used as inputs to the RTM to quantify the impact of dust on SSR. The simulated SSR values using satellite and ground-based aerosol optical properties means that this approach covers all the recognized available aerosol data sources and the RTM outputs that could potentially be used for the proper assessment and corrections of solar power operational loads (Kosmopoulos et al., 2015).

When modeling clear (from clouds) but not clean (from aerosols) sky conditions, aerosol has a particularly important impact on the radiation budget (Schwartz et al., 2014) and hence the AOD, the Angstrom Exponent and the single scattering albedo (SSA) are included in the RTM simulations. The output wavelength range of the radiative fluxes is 285 - 2800 nm in order to facilitate an investigation into the dependence of the irradiance spectrum on particular aerosol parameters for this specific dust case and at the same time to quantify the energy potential. Furthermore, in our RTM simulations, we used the default

aerosol model according to Shettle (1989), the code for spectral irradiance (COSI) developed in 1-Direction for the extraterrestrial solar source spectrum, the parameterization of molecular bands provided by LOWTRAN for the gas absorption, and finally the SDISORT radiative transfer solver (Dahlback & Stamnes, 1991). The RTM simulations were calculated using a band parameterization method based on the correlated-k approximation (Kato et al., 1999) and the





exponential sum fitting technique. This method has been found to be able to offer accurate estimates of the spectral irradiance in spectral intervals comparable with those provided by detailed line-by-line calculations in clear and cloudy sky conditions (Nyamsi et al., 2014).

### 2.2.4 CAMS

Forecast data (1-day-ahead) from Copernicus Atmospheric Monitoring Service (CAMS), based on Monitoring Atmospheric Composition and Climate (MACC) reanalysis tools, were used to provide deeper understanding of the dust transport of the event and also to visualise the plume of dust aerosols. The CAMS data set includes modelling of aerosol and satellite data assimilation. The modelling part is based on ECMWF physical parameterizations concerning aerosol processes and mainly follows the aerosol treatment in the LOA/LMD-Z model (Boucher et al., 2002; Reddy et al., 2005). Detailed description of

the model can be found in Morcrette et al. (2009) and Benedetti et al. (2009). In brief, it estimates dust particles emissions from 10-m wind, soil moisture, albedo in the UV-visible region and land coverage (Morcrette et al., 2008), sea-salt emissions are calculated using a source function (Guelle et al., 2001; Schulz et al., 2004), and other aerosol types emitted by domestic, industrial and transport activities are extracted from SPEW (Speciated Particulate Emission Wizard), and EDGAR (Emission Database for Global Atmospheric Research) annual- or monthly-mean climatologies (Dentener et al., 2006).

Removing aerosols from the atmosphere includes wet and dry decomposition and gravitational settling, and standard schemes for all three of them are used in the model. MODIS AOD at 550nm data are assimilated into a database, applying a bias correction, which uses all available information to determine consistent bias estimates from multiple data sources (Dee and Uppala, 2009). The coupling of these data, provide a database from 2012 at 1hour time steps at 0.4ºX 0.4º spatial resolution.

**3 Results**

**3.1 Synoptic description of the dust event**

As seen in Fig. 1, the establishment of a cold trough during 31 January – 1 February 2015 over West Europe favors the formation of a low pressure system at the Gulf of Genoa. The mean sea level pressure (mslp) reaches 986 hPa over Corsica at 12:00 UTC, 31 January 2015 (Fig. 1a). As the system propagates towards Italy and the Balkans, frontal activity along the

25 North Africa coastline results in increased near surface wind speeds at this area (color scale). Especially over the dust source areas located between the Gulf of Gabes in Tunis and the Gulf of Sirte in Libya, wind speeds at 1000 hPa exceed 15 m s$^{-1}$ and dust particles mobilized are transported over the Mediterranean at the warm sector of the cyclone. The back-trajectories arriving over the Aegean Sea pinpoint to the dust sources as evident by the HYSPLIT 30-hour in Fig. 1b. The air masses arriving at heights 0-2 km over the Aegean Sea at 12:00 UTC, 1 February 2015 originate from Tunis and Libya and are

30 embedded in the cyclonic circulation.



## 3.2 3-D observation

Figure 2 shows the spatial and temporal evolution of the true color imagery (a), cloud fraction (b) and AOD (c) on the days before (top), during (centre) and after (bottom) the peak of the dust event on 01 February 2015. To present the horizontal distributions of MODIS parameters the domain of the eastern Mediterranean Sea is divided into grids of spatial resolution

$0.1^o \times 0.1^o$ deg each, and accordingly the closest MODIS retrieval is assigned to each grid. Note that the days before and after the dust peak were extensively cloud covered (the relationship between AOD and cloud fraction (CF) is described by many researchers e.g. Grandley et al., 2013). Elevated values of AOD (1-3) were observed up to 3 days prior and after the peak (not shown). Based on Fig. 2c centre panel, the dust plume is extends dramatically in the horizontal eventually reaching the Black Sea region with AOD values of similar order but with different lateral characteristics as we will discuss in the

following Sect. 3.4.2. An increase of AOD from the west to the east is observed, reaching maximum values over south Greece. The dust plume yields $AOD_{550nm}$ values larger than 1, while values as high as 3 are persistently evident in the center of the plume. A decreasing gradient to the northeast is also present along the dust plume. Over the larger domain of the western Turkey the high CF and the absence of $AOD_{550nm}$ values prevent the retrieval of the horizontal evolution of the dust plume over Turkey.

Figure 3 shows the 3-D profiling of the dust plume (see Amiridis et al., 2009; 2015; Kosmopoulos et al., 2014; Mamouri et al., 2016). The horizontal distribution is provided by MODIS/Aqua C6 L2 $AOD_{550nm}$ (Left panel) while CALIPSO provides the extinction coefficient profiles at 532nm (Upper Right) and the aerosol subtype classification (Lower Right). The total effect of integrated AOD profile from CAPIPSO as input to the RTM simulations, in terms of SSR output sensitivity and effectiveness in solar energy exploitation, is similar at surface (±5%) compared to the total AOD from MODIS observations

(Kosmopoulos et al., 2014), thus in all RTM simulation cases we used as input the total AOD values from MODIS and CAMS (Sect. 3.4.1). The plume consists mainly of dust aerosol, and polluted dust and dusty marine at the edges, in agreement with similar studies (Gkikas et al., 2016; Papayiannis et al., 2005). Based on the CALIPSO backscatter coefficient and the depolarization ratio (532 nm) profiles, it is possible to decouple the pure dust component from the polluted dust and dusty marine mixtures, hence to estimate the dust contribution to the total AOD (Tesche et al., 2009). According to

CALIPSO the contribution of dust AOD to the total AOD gradually increases from 70% to the south of the dust plume to 100% in the region between $34^o$ and $39.5^o$ latitude. The dust plume extends vertically as high as 3.5 km above sea level. At the edge of the plume (31° N to 33° N), extinction coefficient values at 532 nm are around 0.2 km$^{-1}$, while in the center of the plume (35° N to 40° N), the lidar signal is totally attenuated bellow 1 km. The aerosol extinction coefficient at 532 nm increases from 0.2 km$^{-1}$ at the top of the dust layer, to 2 km$^{-1}$ at about 1.5 km distance from the top of the layer and it reached

values of 10 km$^{-1}$ just before the signal is significantly attenuated.

## 3.3 Ground-based aerosol data



In order to provide a context for the high AOD values observed in the region during the extreme dust incursion on the 1ˢᵗ of February 2015, Fig. 4 presents scatter plots obtained from the multi-sensor aerosol products sampling system (MAPSS: https://giovanni.gsfc.nasa.gov/mapss_explorer/) (Petrenko et al., 2012) of the MODIS/Aqua satellite AOD at 550nm versus coincident ground-based AOD measured by CIMEL sunphotometers in AERONET (Holben et al., 1998). The data presented

5 are outlier-free, spanning the five-year period: 2006-2010 (inclusive) for the entire global record ($\approx$ 50K coincident values) together with the local record at the ATHENS-NOA site ($\approx$ 150 values). The CIMEL at the Athens site has been in operation since 07/04/2008 and provides spectral AOD data at Levels 1.0 (1993 days), 1.5 (1870 days) and 2.0 (1514 days). The coefficient of determination between the satellite and ground-based AOD is high $0.63 \le R^2 \le 0.64$ for Athens (and very high globally: $0.80 \le R^2 \le 0.81$) for both satellite sensors and reflects low root mean-squared errors ($0.07 \le RMSE \le 0.09$). In all

10 cases, a strong high frequency of occurrence peak is apparent in the range $0 \le AOD \le 0.4$. This peak is associated with a modal value of AOD ($\approx$0.16 globally and $\approx$0.13 for Athens). The extreme dust incursion event on the 1 February 2015 is therefore expected to load the aerosol optical depth over the ATHENS-NOA site and be clearly distinguishable from the baseline value of $\approx$0.13.

AERONET ground-based retrieval of Level 1.0 (Version 2) AOD from ATHENS-NOA site, provides approximately hourly

15 records that enable tracing the temporal evolution of the aerosol load (Fig. 5). Values reach up to 2.3-2.5 on 1 February 2015. Level 2.0 data are usually used due to higher quality, and in this particular case, have the same data-points as Level 1.5 data which are provided by the automatic cloud filtering algorithm of AERONET (Smirnov et al., 2000). As shown in Fig. 5, during the period of very high AOD values (before 12 UTC, AOD at 500 nm is higher than 1.9) the cloud screening procedure filtered out about 70% of data points (4 out of 13 passed this test). High AOD values in combination with rapid

20 variation of them lead to the above result. Thus, for studying this dust case, we will use the Level 1.5 products. The cloud-screened inversion data products derived from sky radiance measurements provided by the CIMEL were also used, and both the spectral single scattering albedo (SSA) and the aerosol volume size distribution, key aerosol properties for aerosol classification (see Taylor et al., 2015 and references therein) are presented in Fig. 5. The Level 1.5 SSA presents high values in the near-UV at 470nm that are in excess of 0.85 (rising to $\approx$0.97 in the visible) but importantly, also have extremely low

25 percentage sphericity (0.2-0.6%) as expected for aged dust grains (Dubovik et al., 2002). The typical signature of dust is also demonstrated by the Level 1.5 aerosol volume size distribution which shows a large peak centered on 3-4µm in line with expected microphysical properties of desert dust (Kinne et al., 2003; Taylor et al., 2014).

### 3.4 Impacts of dust on surface solar radiation

### 3.4.1 RTM simulations

30 The intensity of this dust case, was further investigated by compared the induced attenuation in SSR with the mean monthly attenuation percentages over Athens (ATHENS-NOA AERONET site). For this reason we calculated the mean monthly AOD values and ranges based on an 8-years AERONET climatology (07/04/2008-31/03/2016) and the results are shown in



Fig. 6. The range of climatological AODs is ≈ 0.11-0.22 with two peaks in Spring and Summer. Spring is the most favourable season for dust transport from North Africa to Greece (e.g. Kalivitis et al. 2007; Kosmopoulos et al. 2008; Gerasopoulos et al. 2007; 2011), while the summer peak is related to transport of pollution from continental Europe (e.g. Gerasopoulos et al., 2011) and increased agricultural burning and forest fires (e.g. Athanasopoulou et al., 2014). The range

of monthly minimum and maximum AOD values revealed two different peaks in Winter (2.36) and Autumn (1.63) something that indicates that the most extreme aerosol events occur in these two seasons. This finding has to do mainly with intense dust transport in February (our dust case with AOD=1.21-2.36 and median value 1.71) and urban/industrial aerosols (Fameli et al., 2015; Kalivitis et al., 2007; Kosmopoulos et al., 2008), as well as wood burning cases with direct relations to the Greek economic crisis the last 7 years (Vrekoussis et al., 2013). These values were incorporated into the RTM and we

simulated the GHI and DNI percentage attenuation at 40 and 60 degrees of solar zenith angle which are typical solar elevation angles for the region of Greece in the winter season. The results show significantly higher attenuation values for larger SZAs and for DNI in general. In particular, at SZA of 40 degrees, the percentage decrease for GHI is around -5% and for DNI -17%, while at 60 degrees of SZA, the corresponding values are -7% and -25% respectively. All the above results are comparable with similar studies (Papadimas et al., 2012; Tumock et al., 2015; Qian et al., 2007), in terms of AOD range

(0.11-0.25 in EM) and mean aerosol radiative forcing under cloudless conditions (≈ -5% for GHI and -15% for DNI). Under the impact of the studied dust event as measured by AERONET, with median AOD in the specific site around 1.71, the GHI decrease is -37% at $40^{o}$ and -49% at $60^{o}$, and the DNI is -80% and -90% at 40 and $60^{o}$ respectively.

In Fig. 7 we simulated the radiative transfer (RT) using the AERONET station in Athens, in order to highlight the temporal variation of this dust case in 1-hour resolution, as well as the mean dust impact on the solar spectrum. In Fig. 7a we show the

normalized spectral impact on the GHI for the region 285-1050 nm as spectral ratio of the irradiance under actual aerosol conditions to that under aerosol-free conditions. The higher spectral effect is found in the UVA region (around 400 nm) with attenuation of the order of 68%, while in the visible (400-700 nm) and the infrared regions the attenuation is almost 60% and 54%, respectively. These results are similar with relevant studies at various weather and atmospheric conditions (Dimberger et al., 2015; Ishii et al., 2013). In Fig. 7b the corresponding integrated ratios for GHI and DNI are shown. The stronger effect

on DNI as compared to the GHI is apparent, which in Fig. 7b is depicted with DNI ratio values close to zero (absolute blocking) for the entire duration of the day.

In Fig. 8 the effect of dust is shown for various spectral integrals and quantities over Athens for the 1st of February 2015. These estimates were derived from RTM calculations based on actual aerosol conditions (AERONET data in Athens) and dust-free conditions. Specifically we show the diurnal course of DNI, GHI, UV-Index and irradiance in the Visible (VIS),

together with the percentage attenuation for all occasions. Under dust-free conditions, the DNI ranges from 450 to 230 W/m², the GHI from 500 to 270 W/m², the VIS irradiance from 270 to 150 W/m² and the UV-Index from 2.6 to 0.8. Under the dust conditions, the highest values appeared in GHI (250-120 W/m²) and the lowest in DNI (40-10 W/m²) indicating the strong effect of dust on the direct component of SSR. These patterns are reflected in the percentage attenuation as well, with





mean attenuation of -93% in DNI, followed by UV-Index (-70%), VIS (≈ -57%) and GHI (-53%) which is the minimum attenuation compared to all the other SSR fluxes.

Taking into account the significance of being able to forecast expected drastic reduction in SSR, as in the case of the particular dust plume, for e.g. CSP or PV installation management (PVs exploit the GHI and CSPs the DNI), we evaluated the CAMS' forecast from both the total and dust AOD at 550nm found in the dataset described in Sect. 2.2.4, in terms of spatial and quantitative characteristics (Fig. 9). Figure 10 presents the RTM simulations using the CAMS AOD, as well as a direct comparison between the MODIS derived and simulated results with the CAMS 1 day-ahead forecasts. At Fig. 10a we present the MODIS level 3 and CAMS AODs at 550 nm in order to identify the observed (MODIS) and simulated/forecasted (CAMS) dust plume distribution, extent and AOD value intensity. The CAMS simulation follows the MODIS's observed dust plume extent, approaching its distribution but underestimating the peak AOD values (max MODIS values ≈ 3.5 and CAMS ≈ 1.9 over the Greek region). This underestimation pattern of aerosol direct radiative forcing, is a consequence of imperfect forecasted meteorology and fading impact of the initial assimilation of MODIS AOD info on CAMS performance (MACC, 2015; Allen et al., 2013). Yet, despite this difference, the impact on the energy and SSR simulations is of the order of 10% (see below description) in most cases (>90% of the spatial coverage), which highlights that CAMS 1-day ahead forecasts are really of great value and usefulness for solar energy potential planning and policies (Langerock et al., 2015; Charabi and Gastli, 2015; Kosmopoulos et al., 2015; 2017; Ruiz-Arias et al., 2016). At Fig. 10b we simulated GHI (upper) and DNI (lower) under aerosol-free and under MODIS and CAMS aerosol conditions, near local-noon. In all cases we applied smoothing techniques in terms of data fitting to contour lines for better visualization results. The SSR simulations were calculated with the impact of the dust as characterized in terms of high AOD values from MODIS level 3 values and CAMS 1-day ahead forecast. The retrieved AOD for the RTM calculations is at 550 nm, with spatial resolution of 1x1 and 0.4x0.4 degrees (MODIS and CAMS, respectively). The temporal resolution of MODIS overpass imaging is about 1 per day while for CAMS simulation is 1 per hour, highlighting also the ability of CAMS to provide significant information on the temporal evolution of solar energy availability. The panels in Fig. 10c describe the impact on energy in terms of percentage attenuation of SSR indicating the radiative impact of the dust plume over Greece. The simulated results showed mean GHI values of about 500 W/m$^2$ for aerosol-free conditions, while for full aerosol conditions this value is reduced to about 300 W/m$^2$. The corresponding radiation values of DNI are 450 W/m$^2$ for clean and clear sky and around 80 W/m$^2$ for dust event conditions. We need to highlight as well that the maximum AOD that was simulated with the RTM was of the order of 3.5, which classifies this dust event as one of the most intense cases in the eastern Mediterranean. In general, spring presents the higher frequency of dust events (Gerasopoulos et al., 2007; 2011; Gkikas et al., 2012; Kosmopoulos et al., 2008; 2017), while in winter occur the more intense dust events (Kalivitis et al., 2007). The percentage impact of the plume (Fig. 10c) is in the range 30-70% (MODIS) and 30-60% (CAMS) for the GHI and 70-100% and 60-90% (MODIS and CAMS, respectively) for DNI, highlighting and illustrating convincingly the extreme attenuation of the direct component of the total SSR and at the same time quantifying the energy exploitation losses for PV and CSP applications. Overall, concerning the GHI and DNI percentage differences for the MODIS- and CAMS-based RTM simulation, we found that the CAMS forecasts





overestimate the SSR values under high aerosol loads, indicating the limited ability of MACC to predict high AODs (MACC, 2015), while it can efficiently capture the dust plume extent and distribution. As a result, higher percentage differences on DNI following the highest AOD values and the lowest SSR values with minimum induced energy impact (DNI < 50 W/m$^2$) is found. The percentage differences for GHI reach 80-100% for highest AOD values as well, with mean representative GHI attenuation below 50% (see Fig. 10c), highlighting the usefulness for energy forecasting needs and applications.

### 3.4.2 COSMO-ART simulations

Figure 11a depicts the event as captured by the COSMO-ART model application. In particular, the spatial distribution of the total AOD values (at 550nm, 12:00 UTC) is comparable with the respective satellite (MODIS) retrieval. Peak values are simulated over the Aegean Sea and they are higher (AOD = 5.2) compared to the satellite ones (AOD = 3.5). This is partly related to the higher spatial resolution of the model run (0.25°) in comparison to the satellite image (1°). Nevertheless, uncertainties in the dust source functions within aerosol models are common, and are usually treated by model tuning with respect to observations (e.g. Vogel et al., 2006), but this is outside the scope of the current study.

The implications of the interaction between the African dust plume and meteorology, are selectively shown in Fig. 11 (b and c plot). In particular, the incoming solar radiation in terms of GHI and DNI are examined, and their response to the dust plume during the daytime period (mean maximum AOD value ~ 3.5) is shown. As expected, the spatial pattern of the reduced solar energy that reaches the surface resembles that of the dust plume (Fig. 11a), because of the scattering and absorption of the incoming solar radiation by the dust particles. This decrease is more pronounced for the DNI, reaching values up to -180 Wm$^{-2}$, while the effect on the diffuse solar radiation is less intense (up to -100 Wm$^{-2}$, not shown). The effect on the GHI at the surface below the dust plume ranges between –200 Wm$^{-2}$ (dust core) and –20 Wm$^{-2}$ (dust edges), implying that the usage of solar energy in these areas (Cyclades and Crete) is greatly affected during the severe dust storm. This finding is comparable with values estimated in Rémy et. al. (2015), i.e. an effect of -300 Wm$^{-2}$ at the heart of a dust storm over NE Africa (AOD ~ 3). The radiative effect of another dust storm over Western Europe (Bangert et al., 2012) was found smaller, as expected due to the low AOD values (up to 0.5). Considering the radiative efficiency (aerosol radiative effect per unit aerosol optical depth), our findings (60 Wm$^{-2}$) are smaller than those of Rémy et al. (2015) and Stanelle et al. (2010), but this is once more expected, as their findings correspond to hourly (noon) values (-140 to -150 Wm$^{-2}$), experienced over Africa.

Surprisingly, further inland in the Balkan peninsula, where the surface is less affected by the dust plume (AOD < 0.5, dust fraction < 0.4, Fig. 12a) increased amounts of DNI (Fig. 11c) are received, during the dust event. This means that the interaction of dust particles with the atmosphere leads to a positive feedback on solar radiation at the area north of the plume. It is found that the plume tongue is shifted towards the south when the aerosol-meteorology interaction is switched on (base-case run) compared to the scenario with no interaction (not shown). This is related to an increase of the air temperature within the plume (from surface to 2.5 km; not shown), creating a temperature gradient, which leads to a secondary



atmospheric circulation towards the south. The implications on the available solar energy seem important, as the increase in the GHI (and DNI) is found +40 (+60) W m$^{-2}$ over most of the positively-affected area. This follows the mechanism of thermal gradient creation due to the dust radiative effect discussed in Stanelle et al. (2010).

In order to further examine the spatial gradient of the dust effect on the short-wave radiation during the day, thus the hourly energy potential, the hourly zonal means are calculated and plotted (Fig. 12). The negative impact of the dust plume on the DNI, linked to great losses in solar energy potential for CSP systems, lasts from 06:00 to 15:00 UTC (February, 1) and maximizes at 11:00 UTC (-150 Wm$^{-2}$, average -80Wm$^{-2}$ for the southern part of the domain). This value corresponds to more than 90% losses of DNI, due to the presence of the dust plume over south EM. The impact is limited for the rest of the area, ranging from -30 Wm$^{-2}$ (maximum effect over the middle domain at 09:00 UTC) to +25 Wm$^{-2}$ (maximum effect over the northern domain at 11:00 UTC). Nevertheless, percentage values are not negligible, reaching e.g. a dust impact on the DNI of +30% at the northern part of the domain and at 11:00 UTC. Similar is the dust impact on the GHI, referring though to greater absolute values (e.g. -400 Wm$^{-2}$ from the base-case run over the southern domain at 10:00 and 11:00 UTC and -150Wm$^{-2}$ from the scenario which does not take into account the interactions between dust particles and radiation). The consequent percentage losses of the solar energy potential for the PV systems reach 60% (maximum fraction, southern domain).

## 4. Summary and conclusions

This study reconfirms and quantifies high dust aerosol load impact on surface solar radiation. Understanding, determining the range of this impact on attenuation of the surface solar radiation, and providing timely and with adequate accuracy forecast, has major application to emerging solar energy exploitation application. We firstly mapped and studied the 3-D structure of a severe dust event occuring on 1 Feb 2015, via synergy of MODIS/Aqua and CALIOP/CALIPSO space-borne observations. The pure dust outflow yielded AOD$_{550nm}$ values higher than 3 in the plume center of mass, while extinction coefficient values were consistently larger than 0.2 km$^{-1}$ and as high as 10 km$^{-1}$. Then RTM simulations were performed, using MODIS, CAMS and AERONET aerosol data as input. We found GHI values near local noon and under the dust plume of the order of 250 W/m$^2$ and 70 W/m$^2$, for DNI, while the simulated attenuation due to dust was on average -50% for GHI and -90% for DNI (below the dust plume), indicating the effective radiative influence of the dust particles during extreme dust cases. Under cloudless conditions aerosol plays a very important role. For example, the aerosol mean effect in Athens for AOD between 0.11 and 0.22 and for SZA 40$^o$ is 5% and 17% for GHI and DNI respectively, while in the presence of dust (i.e. 1 February 2015, AOD=1.71) the mean aerosol effect is increased, the order of 37% and 80% for GHI and DNI. At larger SZAs (e.g. 60$^o$) this effect can reach values greater than 90%. CAMS provides AOD forecasts enabling forecasting solar energy and the aerosol impact. In this line, we highlighted the usefulness and accuracy (10% as compared with MODIS) of this aerosol forecast data stream, while using COSMO-ART we were able to quantify the important radiative impact of the dust plume over the EM (maximum daily value of the global radiative cooling over Crete is found 200 Wm$^{-2}$)



and the respective mean energy losses for PV and CSP installations, which are consistent with the RTM simulations (≈40% and 80% for GHI and DNI respectively). Since satellite based observations and modeling results become more and easier accessible from year to year, their synergy is promising towards nowcasting, forecasting or analyzing past aerosol events, at spatial resolutions that surface based measurements cannot achieve. The synergistic use of satellite, ground-based

5     measurements and multi-faceted modeling techniques (RTM, CTM and CAMS) in this study, demonstrates the efficiency of such an approach in capturing the impact of dust storms (and expanding of this on other similar aerosol plumes) on SSR and of increasing our understanding of the Earth's radiation budget.

## Acknowledgements

This research has been partly funded by the H2020 GEO-CRADLE project under grant agreement No 690133. We also

10    acknowledge ACTRIS-2, which has received funding from EU H2020 research and innovation programme (grant agreement No 654109).

## Nomenclature & abbreviations

| | |
|---|---|
| AERONET | Aerosol Robotic Network |
| AF | Actinic Flux |
| AOD | Aerosol Optical Depth |
| ART | Aerosol and Reactive Trace gases |
| CAPIOL | Cloud Aerosol Lidar with Orthogonal Polarization |
| CALIPSO | Cloud-Aerosol Lidar and Infrared Pathfinder Satellite Observations |
| CAMS | Copernicus Atmosphere Monitoring Service |
| CF | Cloud Fraction |
| CSP | Concentrated Solar Plants |
| CTM | Chemical Transport Model |
| C6 | Collection 6 |
| DB | Deep Blue |
| DNI | Direct Normal Irradiance |
| DT | Dark Target |
| EARLINET | European Aerosol Lidar Network |
| ECMWF | European Centre for Medium-Range Weather Forecasts |
| EDGAR | Emission Database for Global Atmospheric Research |
| EM | Eastern Mediterranean |
| FNL | Final Analysis Data |
| GAW | Global Atmosphere Watch |
| GHI | Global Horizontal Irradiance |
| GOCART | Goddard Chemistry Aerosol Radiation and Transport |
| MACC | Monitoring Atmospheric Composition and Climate |
| MAPSS | Multi-sensor Aerosol Products Sampling System |
| MODIS | MODerate resolution Imaging Spectroradiometer |
| NCEP | National Centers for Environmental Protection |
| PV | Photovoltaics |



| RTM | Radiative Transfer Model |
|---|---|
| SSR | Surface Solar Radiation |
| SPEW | Speciated Particulate Emission Wizard |
| SSA | Single Scattering Albedo |
| SW | Shortwave |
| SZA | Solar Zenith Angle |
| TOA | Top of Atmosphere |
| UV | Ultraviolet |
| VIS | Visible |

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





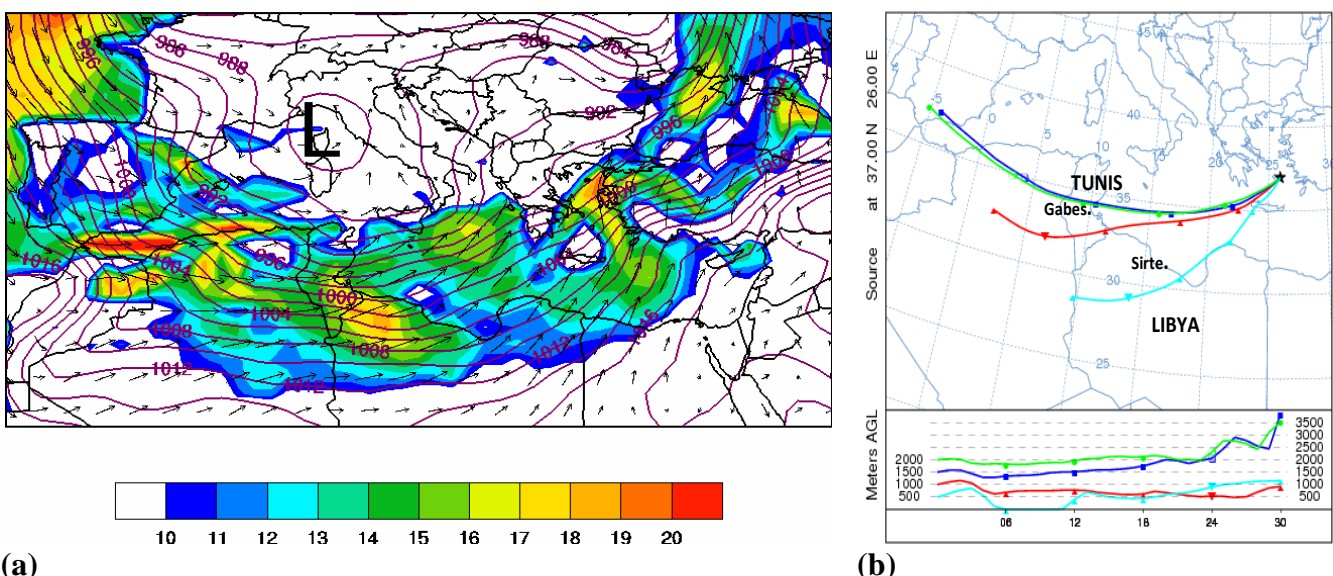

(a)  (b)

**Figure 1: NCEP final analysis data (FNL) at 12:00 UTC, 31 January 2015 (a). The wind speed > 10 ms⁻¹ at 1000 hPa is shown (color scale and vectors) overlaid with mean sea level pressure (contour lines). The low pressure center is denoted with L. The**
15  **HYSPLIT back-trajectories arriving over the Aegean Sea at 12:00 UTC, 1 February 2015 (b).**





**(a)**          **(b)**          **(c)**

**Figure 2: Satellite observation of the temporal evolution of the dust incursion peaking on 1 Feb 2015 over Greece from MODIS/Aqua satellite data. (a) True colour imagery based on Bands 1,3 and 4. (b) Evolution of the cloud fraction and (c) AOD on the day before (top), during (centre) and after (bottom) peak dust. Note that the days either side of the peak are extensively cloud covered. High values of AOD were observed up to 3 days prior and after the peak.**



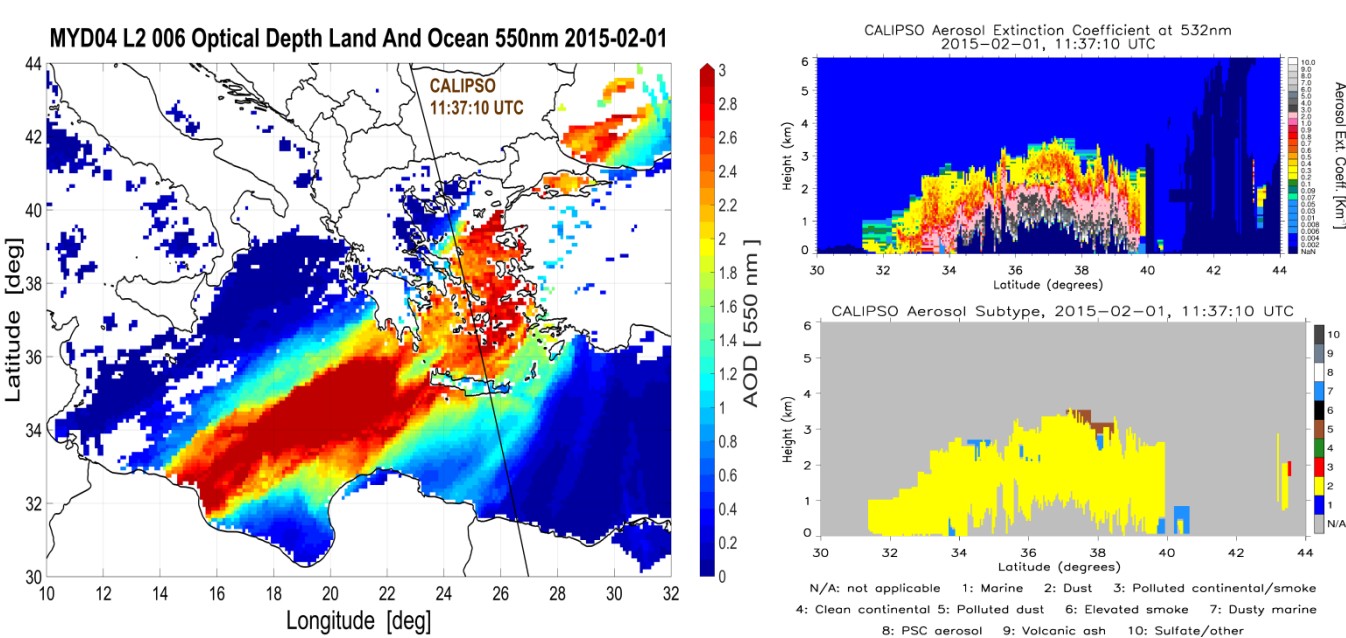

**Figure 3: 3D profiling of the dust incursion on 1 Feb 2015: spatial extent provided by MODIS Level 2 (collection 6) AOD at 550nm (Left) together with CALIPSO profiles at 532nm of the extinction coefficient (Upper Right) and aerosol sub-typing (Lower Right) in the profile taken at 11:37 UTC (indicated by the distinct black-coloured track line in the left panel).**





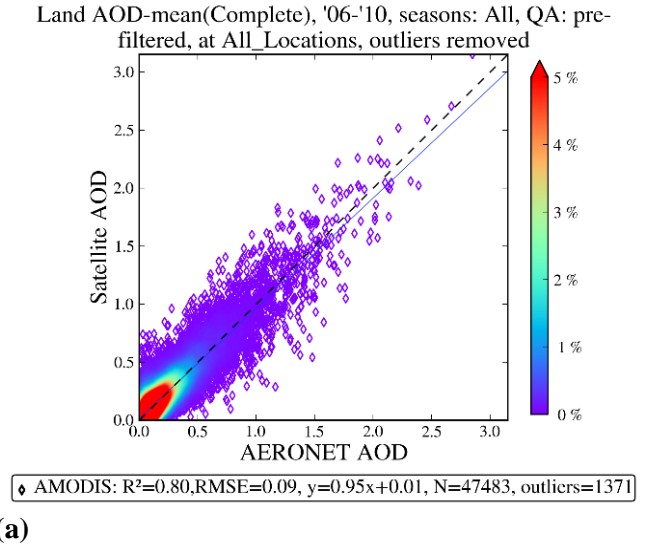

**(a)**                                                    **(b)**

**Figure 4: Scatter plots of satellite AOD (550nm) from MODIS/Aqua versus coincident AOD from AERONET for the five year period 2006-2010 with outliers removed. (a) shows the global record and (b) is for the ATHENS-NOA site. The colour bar describes the percentage frequency of AOD values occurrence (source of the plots: https://giovanni.gsfc.nasa.gov/mapss_explorer).**

35



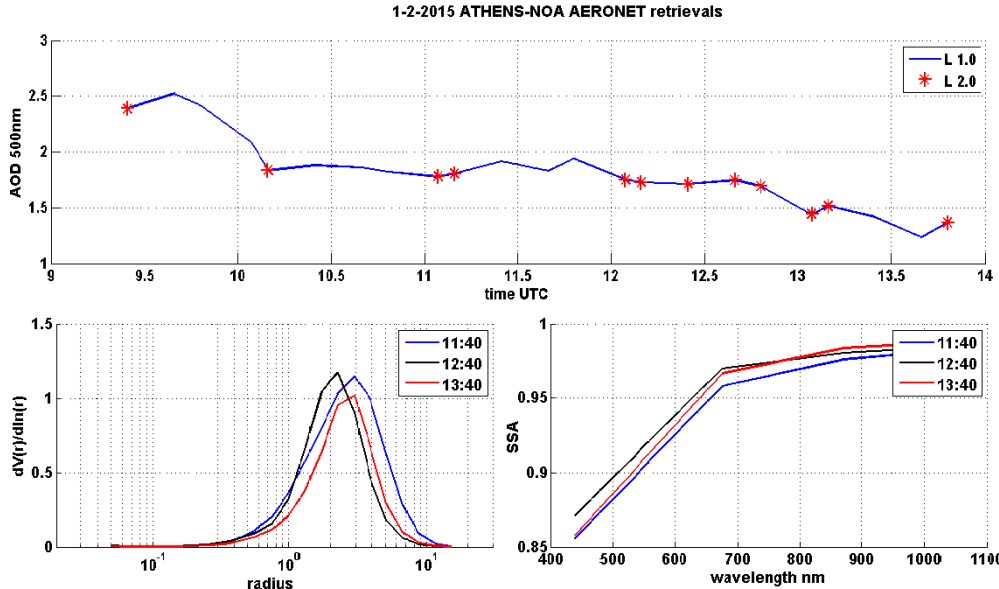

**Figure 5: Temporal evolution of the AOD at 500 nm obtained from the Level 1.0 and 2.0 inversion algorithm of the CIMEL sunphotometer at the Athens site during the day of peak dust incursion over the region on 1 Feb 2015 (upper) together with aerosol volume size distribution (lower left) and the spectral SSA (lower right) retrieved from the Level 1.5 inversion algorithm on the same day.**



**Figure 6: Mean monthly GHI and DNI percentage attenuation in Athens as a function of solar zenith angle and AOD (a) together**
15 **with the mean monthly range of AOD (b), the percentage attenuation range for GHI (c) and DNI (d). The red-shaded insets shows**
**the corresponding median values and ranges at the peak of the strong dust incursion over the region on 1 Feb 2015.**





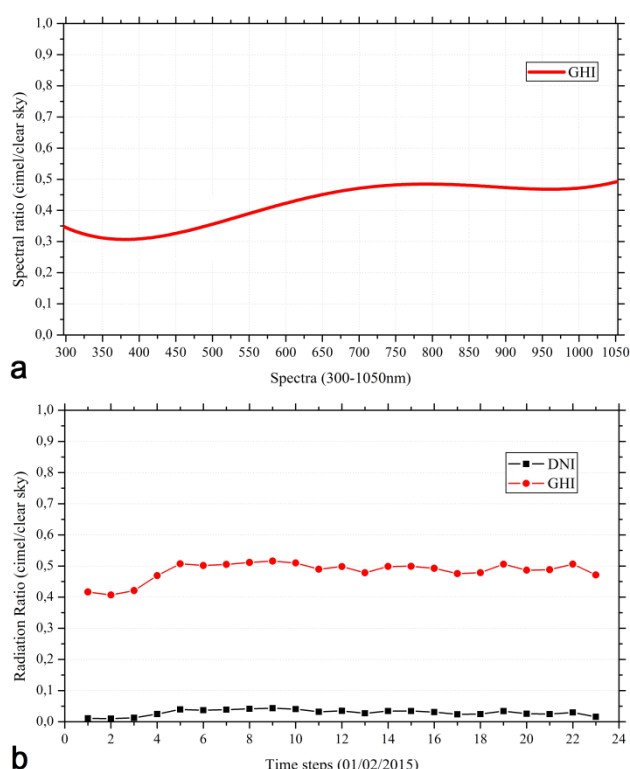

**Figure 7: (a) The spectral effect of the extreme dust event of 1 Feb 2015 at the Athens station as depicted by the ratio of the AERONET Cimel sunphotometer irradiance to the clear sky irradiance calculated with a RTM for the same atmospheric conditions. Note the increased attenuation at shorter wavelengths. (b) Temporal evolution of the hourly value of the ratio for the GHI and DNI output by the RTM.**





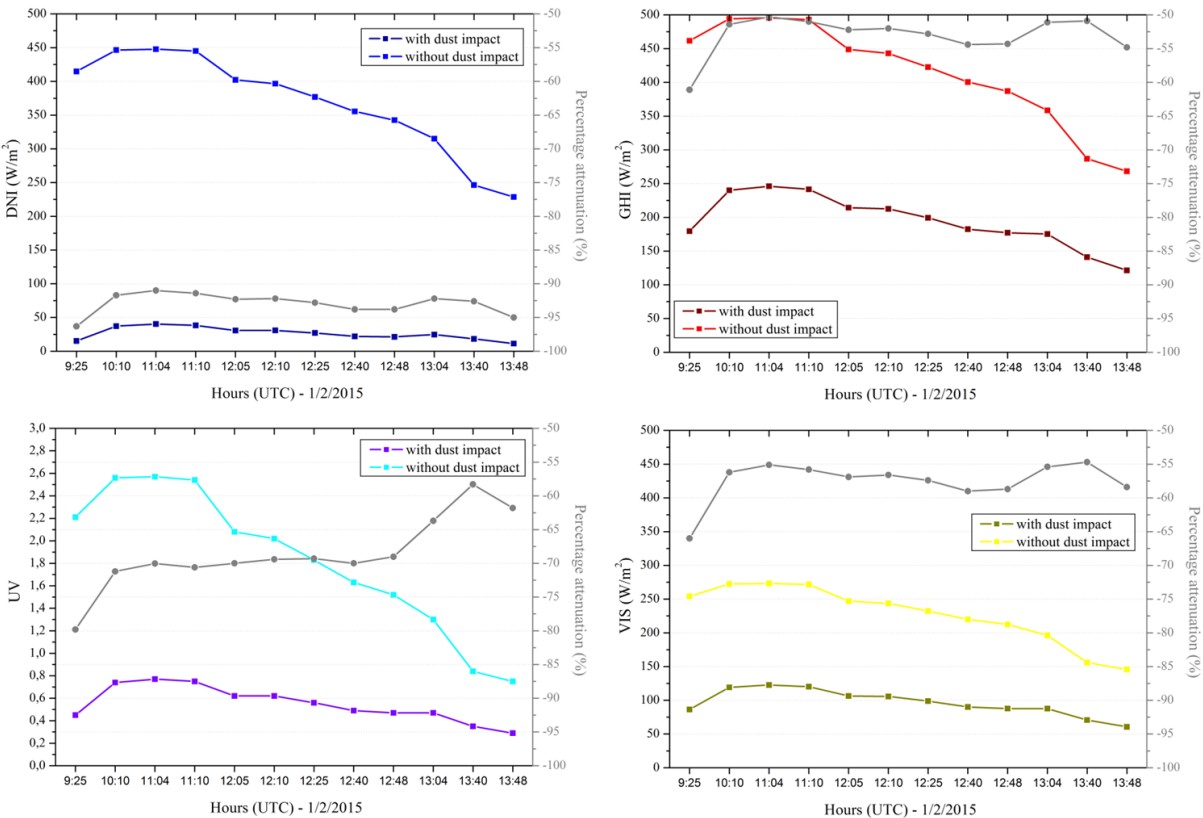

**Figure 8: Temporal evolution of GHI, DNI, UV and VIS irradiances during the extreme dust event of 1 Feb 2015 at the Athens**
**station. The modeled values (without dust impact) coincident with AERONET Cimel measurements (with dust impact) are shown**
**together with the % attenuation due to impact of dust.**

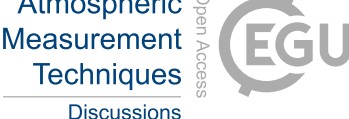

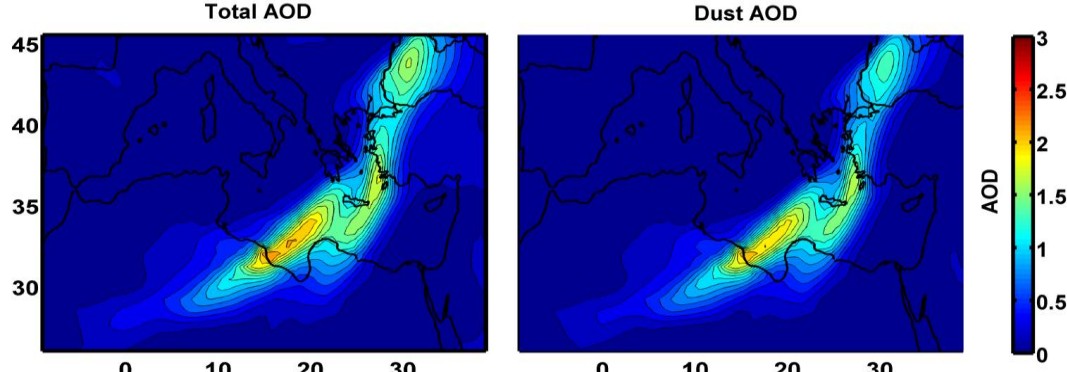

**Figure 9: The spatial extent of total AOD (left) and dust AOD (right) at 550nm provided by CAMS. These maps depict predictions by the MACC one-day-ahead forecast (spatial resolution: 0.4 x 0.4 degrees). The maps correspond to 12:00 UTC, when maximum AOD ≈ 2.3. The temporal resolution of the forecast is 1 hour.**

35

40





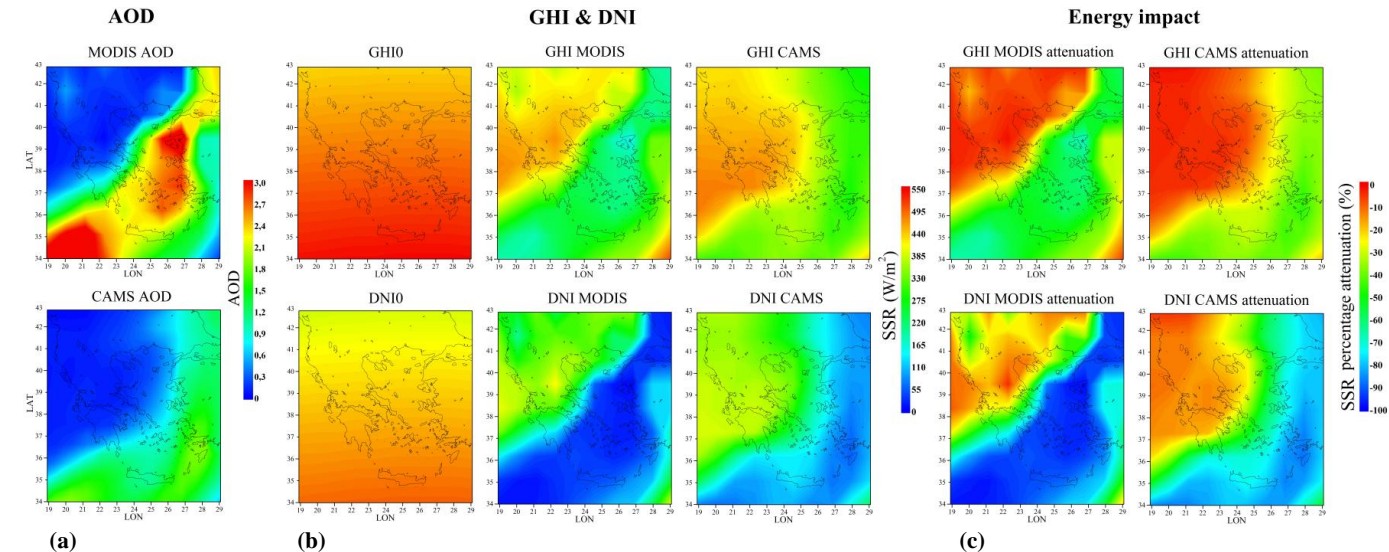

(a)    (b)    (c)

**Figure 10:** (a) AOD from MODIS level 3 and the CAMS 1-day ahead forecast. (b) RTM simulations at local noon on the day of the incursion on 1 Feb 2015 for GHI and DNI. GHI0 and DNI0 represent the simulations without dust (with only the effects of SZA). (c) The energy impact in terms of percentage attenuation relative to the dust-free simulations for GHI and DNI under MODIS- and CAMS-based AODs. For the GHI, the attenuation is about 30 – 70 % for MODIS and about 30 - 60 % for CAMS. For the DNI, the attenuation is about 70 – 100 % for MODIS and about 60 - 90 % for CAMS.





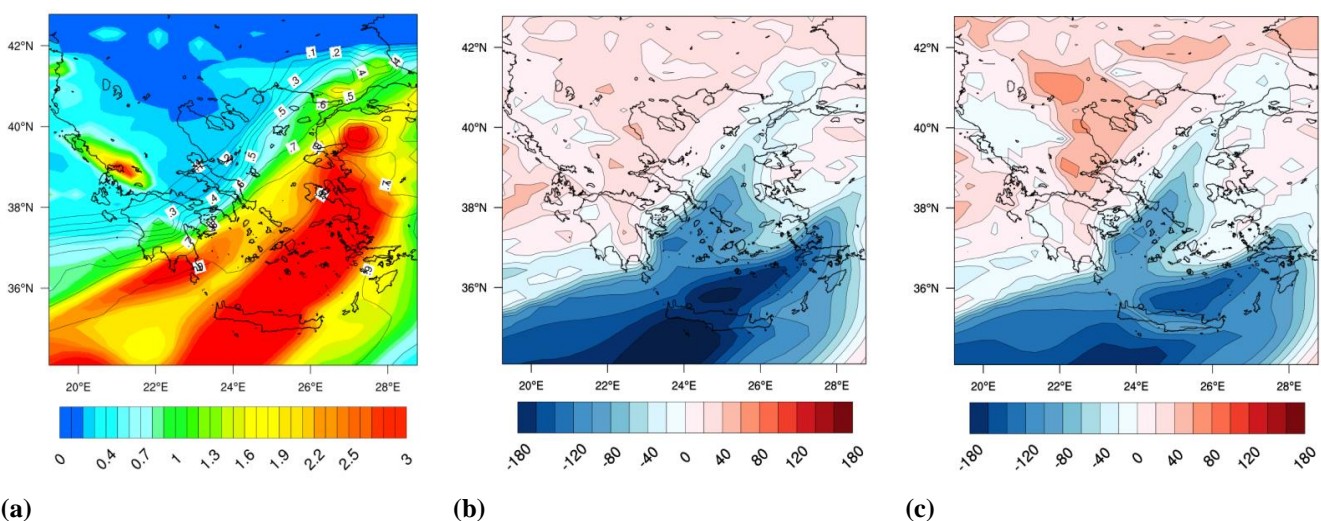

(a)                          (b)                          (c)

20  **Figure 11: The spatial distribution of the: (a) total AOD (at 550nm). Iso-lines indicate the dust fraction of the total AOD model values. The map corresponds to 12:00 UTC, when maximum AOD ≈ 5.2. (b) dust plume effect on GHI and (c) DNI (in W/m²) at surface, averaged for the daytime hours of the extreme dust event of 1 Feb 2015 over Greece as predicted by COSMO-ART.**

35







**Figure 12:** The diurnal variation of the GHI (a, c, e) and DNI (b, d, f) short-wave radiation zonally-averaged over the northern,
middle and southern domains (maps shown). The dark grey line corresponds to the base-case COSMO-ART run that includes
aerosol-meteorology. The light grey line corresponds to the scenario when aerosol-meteorology interaction is switched off.