# Peer review of "Dust impact on surface solar irradiance assessed with model simulations, satellite observations and ground-based measurements"

_Atmospheric Measurement Techniques, 2017_

## Referee Comment (RC1) · Anonymous Referee #1 · 4 May 2017

General comments:

The manuscript describes and analyses a strong episode of dust on the eastern Mediterranean region, by using satellite, surface and models for estimating the impact of dust on surface radiation.

The analysis is sound and merits publication in this journal. It is also of interest for applications such as solar energy forecast. However it would be interesting to add real measurements of solar energy plants to the estimations here described, if available. In any case, only minor comments are asked to be taken into account.

[Figure]

Specific comments:

- Page 2, lines 24-25: please state the reason to choose LibRadTran model among the different models available (if any specific reason). It would be useful to state breafly the advantages. Cite accordingly. - Page 4, line 15: "the temporal resoluiton of AERONET measurements is very high (∼1 per 10 minutes)". This statement is relative. AERONET performs direct AOD measurements broadly every 15 minutes. Other instruments measure the AOD every 1 minute or less. So I do not consider the temporal resolution to be very high in relation to other instruments available. - Page 7, line 18: what is the comparison of CALIPSO and MODIS in terms of AOD for the specific episode? - Page 8, lines 14-27: only level 2 retrievals should be used for climate data records, although level 1.5 are still useful for analysing especific cases. The authors decided to use only level 1.5 data. Do the authors consider that level 2 data criteria were too strict for this particular case, based on their experience or other simultaneous measurements? Could the authors state which AERONET criteria were decisive for not attaining level 2? - Page 8, line 31-32: for the 8 year climatology, level 1.5 or level 2 was used?

Minor corrections: - Page 7, line 8: "extends" or "is extended" - Page 7, line 18: CALIPSO - Page 8, line 30: "by comparing"? - Page 10, line 3: please revise - Page 10, line 6: describe figure 9 before passing on figure 10 - Page 10, line 11: perhaps I missed something, but I would say you refer to AOD instead of radiative forcing. - Figure 5: add axis units - Figure 6: avoid using smoothed lines between points, as in the other figures - Figure 8: plot c, state UV index or units. Same for caption.

---

## Author Comment (AC2) · 12 Jun 2017

We acknowledge the reviewer #2 for the meaningful comments. We have hopefully addressed all the points that were raised, and we are optimistic that after the reviewer's valuable general, specific and technical corrections, the paper has been upgraded. Please find attached a zip archive with the following pdf files: (i) The answers to the reviewer comments, (ii) A correction version of the paper

Please also note the supplement to this comment:

[Figure]

http://www.atmos-meas-tech-discuss.net/amt-2017-79/amt-2017-79-AC2-supplement.zip

---

## Author Response (AR1)

reviewer #1

The manuscript describes and analyses a strong episode of dust on the eastern Mediterranean region, by using satellite, surface and models for estimating the impact of dust on surface radiation. The analysis is sound and merits publication in this journal. It is also of interest for applications such as solar energy forecast. In any case, only minor comments are asked to be taken into account.

■       However it would be interesting to add real measurements of solar energy plants to the estimations here described, if available.

Author's reply: The solar energy plants receive the input energy which is the local energy yield and produce the energy output which is only a percentage of the inputs because of the total system and material losses. The climatological, topographical and geographical conditions are affecting the performance as well. As a result, adding real measurements of solar energy plants is a complex issue while the comparison of RTM simulations against real plants measurements, needs a completely separate and energy-losses analysis. We agree with the reviewer that this would be interesting and we hope to have the opportunity to study real solar plant measurements in a following study.

■       Page 2, lines 24-25: please state the reason to choose LibRadTran model among the different models available (if any specific reason). It would be useful to state briefly the advantages. Cite accordingly.

Author's reply: We thank the reviewer for this comment. In the revised paper we included a brief description of the advantages using libRadtran.

■       Page 4, line 15: "the temporal resolution of AERONET measurements is very high (_1 per 10 minutes)". This statement is relative. AERONET performs direct AOD measurements broadly every 15 minutes. Other instruments measure the AOD every 1 minute or less. So I do not consider the temporal resolution to be very high in relation to other instruments available.

Author's reply: The sentence has been restated.

■       Page 7, line 18: what is the comparison of CALIPSO and MODIS in terms of AOD for the specific episode?

Author's reply: The maximum AOD observed with CALIOP is about 3 in the center of the plume, while the corresponding AOD from MODIS is almost 3.5 over the Greek region and about 3 over the plume part that CALIPSO overpasses. We added the proposed direct comparison of these two sensors in the revised version.

5  ▪  Page 8, lines 14-27: only level 2 retrievals should be used for climate data records, although level 1.5 are still useful for analysing specific cases. The authors decided to use only level 1.5 data. Do the authors consider that level 2 data criteria were too strict for this particular case, based on their experience or other simultaneous measurements? Could the authors state which AERONET criteria were decisive for not attaining level 2?

10  Author's reply: In this particular Dust Case, level 1.5 and level 2.0 AOD products had exactly the same data points. Figure 5 upper plot demonstrated both level 1.0 and level 2.0 (identical to level 1.5). The automatic filtering algorithm (described at Smirnov et al.,2000) filters out a lot of AOD values due to very rapid change of aerosol load. For AOD study we have used level 1.0 data because this high values were considered as "clouds" when in reality there were irregular high dust concentration. There is no way to have this estimation using only CIMEL measurements, so it cannot be applied in

15  AERONET algorithms or generalize this method. But the general picture of the atmospheric conditions on that particular day, described in detail in the present study, drives us to this approach. For inversions retrievals we have used level 1.5 products, although most measurements have already been filtered out by the cloud screening procedure, we just wanted to skip the sza>50º criterion used for level 2.0. We keep in mind that retrievals at sza<50º have higher uncertainties, but considering our scientific interest on that episode and the importance of having some ground based estimation of the nature

20  of aerosols we chose to proceed with level 1.5. Inversion products haven't been used in model calculations, so these higher uncertainties are not spread in our results. The paragraph has been restated to clarify the difference between AOD level 1.0 and inversions level 1.5 products.

▪  Page 8, line 31-32: for the 8 year climatology, level 1.5 or level 2 was used?

Author's reply: The climatology was based on Level 1.5 data. This information was added in the revised version.

▪  Page 7, line 8: "extends" or "is extended"

30  Author's reply: Corrected.

▪  Page 7, line 18: CALIPSO - Page 8, line 30: "by comparing"?

Author's reply: We corrected these two grammatical failures. Thank you for noticing them.

- Page 10, line 3: please revise

Author's reply: We revised the sentence, thank you for mention this.

- Page 10, line 6: describe figure 9 before passing on figure 10

Author's reply: You are right. We corrected that.

- Page 10, line 11: perhaps I missed something, but I would say you refer to AOD instead of radiative forcing.

Author's reply: We want to thank the reviewer for the carefully reading and this conceptual correction. The reference was for the AOD indeed, so we made the appropriate revision.

- Figure 5: add axis units

Author's reply: The axis units were added in the revised paper.

- Figure 6: avoid using smoothed lines between points, as in the other figures

Author's reply: In the revised paper we used straight lines between points in Figure 6 and anywhere else it was applicable (e.g. not in mean spectrally polynomial fitting case).

- Figure 8: plot c, state UV index or units. Same for caption.

Author's reply: We thank the reviewer for this definition omission. We now fixed this issue for both plot and caption.

reviewer #2

This is a case study of a dust event that occurred in the Eastern Mediterranean during 30 January and 3 February 2015, and its impact on the solar radiation received at the surface. Used are observations from a variety of sources such as AERONET, MODIS, CALIPSO, a radiative transfer model and a chemical transport model and the 1-day ahead forecasts from the Copernicus Atmosphere Monitoring Service. It is reported that such a dust event can result in attenuation of the global radiation by as much as 40-50%, and a decrease of 80-90% in the direct component.

The approach used in this study follows methodologies implemented in numerous previous studies on the impact of dust on the reduction of solar radiation at the surface (and/or at the TOA). As such, the results obtained are to be expected. Some references to previous relevant work are included here.

■        The actual value of the analysis to forecasting such impacts is not obvious since in real time, all the information that was available in this case in hindsight, will not be available in real time. Therefore, such impacts on solar energy planning will have to be estimated from previous knowledge of anticipated reduction.

Author's reply: The 1-day ahead CAMS aerosol forecasts can provide the basic model input information in order to predict with accuracy (within 10% as compared to the MODIS hindsight data) the AOD values in an operationally good temporal and spatial resolution (1-hour, 0.4 degrees). The other critical input is the solar zenith angle which can be pre-calculated, so the whole real time approach is feasible in terms of pre-calculated and forecasted input data to the RTM. As a result, the main output can be operational maps (like in Fig. 10) of GHI, DNI and percentage attenuation. We have now added this additional information to the text as to briefly describe the real time possibilities.

■        Since the case was well documented with information from numerous sources (no validation at the surface was attempted), perhaps a brief communication on this case with a substantially reduced number of figures would be appropriate.

Author's reply: This dust case was the most extreme events in the last 5 years for the specific area, so we decided to not only study the energy impact which is the main scope of this paper, but to investigate from an observational point of view its intensity and characteristics. We believe that this complementary approach was highlighted throughout the text.

■        The bibliography provided is very selective. Some relevant publications:

a. Tegen, Ina, Lacis, Andrew A., Fung, Inez Nature, 1996. The influence on climate forcing of mineral aerosols from disturbed soils. Apr 4, 380, 6573, ProQuest pg. 419.

b. Li, F, Vogelmann AM, Ramanathan V., 2004. Saharan dust aerosol radiative forcing measured from space. Journal of Climate. 17:2558-2571.

c. Diaz, J. P., F. J. Exposito, J. Torres, F. Herrera, J. M. Prospero, and M. C. Romero, 2001. Radiative properties of aerosols in Saharan dust outbreaks using ground-based and satellite data: Applications to radiative forcing. J. Geophys. Res., 106, 18 403– 18 416.

d. Haywood, J. M., P. N. Francis, M. D. Glew, and J. P. Taylor, 2001. Optical properties and direct radiative effect of Sharan
5  dust: A case study of two Saharan dust outbreaks using aircraft data. J. Geophys. Res., 106, 18 417–18 430.

e. Kaufman, Y. J., A. Karnieli, and D. Tanre, 2000. Detection of dust over the desert by EOS-MODIS. IEEE Trans. Geosci. Remote Sens., 38, 525–531.

f. Kaufman, Y.J., D. Tanre, O. Dubovik, A. Karnieli, and L. A. Remer, 2001. Absorption of sunlight by dust as inferred from satellite and ground-based remote sensing. Geophys. Res. Lett., 28, 1479– 1482.

10  g. Pandithurai, G., et al., 2008. Aerosol radiative forcing during dust events over New Delhi, India, J. Geophys. Res., doi: 10.1029/2008JD009804.

h. Miller, R. L., I. Tegen, and J. Perlwitz, 2004. Surface radiative forcing by soil dust aerosols and the hydrologic cycle, J. Geophys. Res., 109, D04203, doi: 10.1029/2003JD004085.

15  Author's reply: We thank the reviewer for these additional references. It will be a valuable addition, so we included them in the revised paper.

▪  When providing information on outliers, indicate the % of total number of points available.

20  Author's reply: The outliers represent less than 0.5% of the data (250 points) of total coincident values. We thank the reviewer for mentioning this omission. We added this information in the revised version.

▪  Since the paper deals with the Eastern Mediterranean, examples of CSP installation in that region should be mentioned (instead of a facility in Western Med).
25
Author's reply: We added some relevant CSP installations for the Eastern Mediterranean as well since the whole Mediterranean region is often affected by Saharan dust plumes.

▪  Acronym and references should be provided the first time used (e.g., P. 2, L. 25: libRadtran).
30
Author's reply: We want to thank the reviewers for all the careful remarks. We have now corrected all the structural issues.

▪ P. 3, L. 23, stated: "In this paper MODIS Aqua C6 L2 is used". There are differences between Terra and Aqua. Why Aqua was selected?

Author's reply: The present study investigates the impact of dust aerosols during an extreme dust event on surface solar radiation. In the framework of the study and in order to describe the dust event both passive and active satellite remote sensing instrumentation (MODIS and CALIOP) are utilized. Indeed the reviewer is right, the anomalously high aerosol load recorded between the 30th of January and the 3rd of February 2015 over the domain of eastern Mediterranean Sea is captured both by Aqua MODIS and Terra MODIS. There are differences between Terra and Aqua. Both Terra and Aqua are sun-synchronous, near-polar circular orbit satellites. Terra crosses in descending node the equator at approximately 10:30 A.M. local time. On the contrary, Aqua crosses in ascending node the equator at approximately 1:30 P.M. local time. Both Aqua-MODIS and Terra-MODIS spectroradiometers image the same domain on the Earth's surface with a difference of approximately three hours. The dust event, having a duration of three days, is captured well by both MODIS sensors. The selection of Aqua-MODIS in the present study is related to the orbit of Aqua which meets the needs of providing the 3-dimentional overview and description of the dust event. Aqua, being part of the A-Train constellation of earth observation satellites, flies in formation with CALIPSO. Consequently, the synergy of Aqua-MODIS and CALIPSO-CALIOP results in the horizontal and vertical description of the dust event, hence on the 3-dimensional overview. This could not have been achieved if focusing on the Terra-MODIS retrievals. For this clarification, we added the above brief description in the revised paper.

▪ P. 4, L. 19, stated: "The final analysis data (FNL) of the National Center for Environmental Protection (NCEP) are used for the assessment of the meteorological conditions". Later on, on P. 4, L. 27: Stated: "COSMO-ART is a regional atmospheric model which couples online meteorology and chemistry and is used". Some explanation is needed why both are needed.

Author's reply: The FNL are meteorological reanalysis data at 1x1 resolution and are used to represent the synoptic scale conditions during this period. Also FNL is part of the GDAS data assimilation system that we used to drive the HYSPLIT runs and as seen in Figures 1a,b both datasets confirm the favorable conditions for the transport of Saharan dust towards the Aegean.

On the other hand, COSMO-ART is a more complex prognostic model and it is used for the numerical simulation of the interactions between atmospheric chemistry and meteorology at regional scale. The model solves all relevant equations at every time step (30 sec), so that the aerosol (in our case, dust particles) effects on the earth radiative budget are estimated accordingly (e.g. Figures 11 and 12).

▪ P. 11, L. 28, stated: "Surprisingly, further inland in the Balkan peninsula, where the surface is less affected by the dust plume during the dust event. This means that the interaction of dust particles with the atmosphere leads to a positive feedback on solar radiation at the area north of the plume." The connection between these two statements requires clarification.

Author's reply: We realize that the connection between the position of the dust plume tongue and the direct radiative effects is not evident, because of the structure of the relevant paragraph. In short –and as already described in previous relevant studies (e.g. Stanelle et al., 2010), the interactions between dust aerosol and radiation creates a thermal gradient in the atmosphere, which causes a shift of the dust tongue towards the south. In other words, when the interaction is on (base-case

10 run) more amounts of direct solar radiation are received at ground further inland the Balkan peninsula, due to the aforementioned mobility of the dust plume from north to south. In order that this is clearer in the revised document, a restructuring in the relevant paragraph is performed.

▪ P. 12, L.17, stated: "This study reconfirms and quantifies high dust aerosol load impact on surface solar radiation."

15 Since this study only "reconfirms" what is already known, it is recommended to condense it to a short communication.

Author's reply: We thank the reviewer for this comment in order to clarify the significance of the communication followed in this paper. Actually, by concluding in this way we wanted to "reconfirm" such quantified aerosol impacts on SSR, adding a perspective on operational energy planning using a synergy of CAMS AOD forecasts and pre-calculated inputs like the

20 SZA. This knowledge will enable the evolution of accurate near real time, nowcasting and forecasting SSR estimation models and systems as to provide already know information at the right time and not as post processing reanalysis data. At the same time, since this dust case was one of the most intense of last 5 years, it is useful for the readers to follow our proposed methodology with reference to a recent case study. As a result, we highlighted the multifaceted approach covering the model simulations from RTM and CTM in conjunction with the predicted AOD from CAMS, its validation against

25 MODIS data and the 3D observational dust plume description. We strongly believe that this study forwards the quantification of the PV and CSP losses from dust events and it will be a step to the right direction of the energy related policies.

Authors: Once again, we thank the reviewers for their constructive comments and we believe that after the proposed

30 revisions this study was overall upgraded.

[revised manuscript text omitted]